Using GIS to examine biogeographic and macroevolutionary patterns in some late Paleozoic cephalopods from the North American Midcontinent Sea

Kolis Kayla M. 1
Lieberman Bruce S. blieber@ku.edu 1 2
1 Biodiversity Institute, University of Kansas , Lawrence , KS , United States of America
2 Department of Ecology & Evolutionary Biology, University of Kansas , Lawrence , KS , United States of America
Piñeiro Graciela
Electronic publication date: 2019 May 13
Publication date: 2019
Volume: 7
Electronic Location ID: e6910
Received 2018 Sep 17; Accepted 2019 Mar 23
Copyright: ©2019 Kolis and Lieberman
Copyright year: 2019
Copyright holder: Kolis and Lieberman
License: This is an open access article distributed under the terms of the Creative Commons Attribution License, which permits unrestricted use, distribution, reproduction and adaptation in any medium and for any purpose provided that it is properly attributed. For attribution, the original author(s), title, publication source (PeerJ) and either DOI or URL of the article must be cited.
License URL: https://creativecommons.org/licenses/by/4.0/

Keywords: Geographic information systems (GIS), Macroevolution, Late paleozoic, Cephalopods, Biogeography

Funding: US National Science Foundation (NSF) Emerging Frontiers (EF) 1206757 NSF Division of Biological Infrastructure (DBI) 1602067 This research was supported by US National Science Foundation (NSF) Emerging Frontiers (EF) grant 1206757, NSF Division of Biological Infrastructure (DBI) grant 1602067, and a Panorama grant from the Biodiversity Institute, University of Kansas. The funders had no role in study design, data collection and analysis, decision to publish, or preparation of the manuscript.

==============================
Geographic range is an important macroevolutionary parameter frequently considered in paleontological studies as species’ distributions and range sizes are determined by a variety of biotic and abiotic factors well known to affect the differential birth and death of species. Thus, considering how distributions and range sizes fluctuate over time can provide important insight into evolutionary dynamics. This study uses Geographic Information Systems (GIS) and analyses of evolutionary rates to examine how in some species within the Cephalopoda, an important pelagic clade, geographic range size and rates of speciation and extinction changed throughout the Pennsylvanian and early Permian in the North American Midcontinent Sea. This period is particularly interesting for biogeographic and evolutionary studies because it is characterized by repetitive interglacial-glacial cycles, a global transition from an icehouse to a greenhouse climate during the Late Paleozoic Ice Age, and decelerated macroevolutionary dynamics, i.e. low speciation and extinction rates. The analyses presented herein indicate that cephalopod species diversity was not completely static and actually fluctuated throughout the Pennsylvanian and early Permian, matching findings from other studies. However, contrary to some other studies, the mean geographic ranges of cephalopod species did not change significantly through time, despite numerous climate oscillations; further, geographic range size did not correlate with rates of speciation and extinction. These results suggest that pelagic organisms may have responded differently to late Paleozoic climate changes than benthic organisms, although additional consideration of this issue is needed. Finally, these results indicate that, at least in the case of cephalopods, macroevolution during the late Paleozoic was more dynamic than previously characterized, and patterns may have varied across different clades during this interval.

Introduction

Much work has focused on the relationship between geographic range size and rates of speciation and extinction (e.g.,  Vrba, 1980; Jablonski, 1986; Eldredge, 1989; Stanley, 1990; Lieberman, 2000; Jablonski & Roy, 2003; Rode & Lieberman, 2004; Rode & Lieberman, 2005; Kiessling & Aberhan, 2007; Liow, 2007; Payne & Finnegan, 2007; Abe & Lieberman, 2009; Stigall, 2010; Myers & Saupe, 2013; Myers, MacKenzie & Lieberman, 2013; Dunhill & Wills, 2015; Jablonski & Hunt, 2015; Orzechowski et al., 2015; Saupe et al., 2015; Castiglione et al., 2017; Pie & Meyer, 2017; Simões et al., 2016; Lam, Stigall & Matzke, 2018; Schneider, 2018). Furthermore, the use of Geographic Information Systems (GIS) has greatly facilitated investigations into this macroevolutionary relationship (Stigall & Lieberman, 2006; Hendricks, Lieberman & Stigall, 2008; Dunhill, 2012; Myers, MacKenzie & Lieberman, 2013; Dunhill & Wills, 2015; Lieberman & Kimmig, 2018). Here, we focus on how geographic range size and rates of speciation and extinction changed throughout the Pennsylvanian and early Permian in the North American Midcontinent Sea in the Cephalopoda, an important clade of pelagic invertebrates (Kullmann, 1983; Kullmann, 1985; House, 1985; Becker & Kullmann, 1996; Landman, Tanabe & Davis, 1996; Wiedmann & Kullmann, 1996; Monnet, De Baets & Klug, 2011; Korn & Klug, 2012; Klug et al., 2015; Korn, Klug & Walton, 2015), using GIS. This time interval is particularly interesting for biogeographic and evolutionary analysis because it is characterized by repetitive glacial-interglacial cycles, and a global transition from an icehouse to greenhouse climate during the Late Paleozoic Ice Age (LPIA) (Montañez & Poulsen, 2013). Further, it is generally considered a time of sluggish macroevolutionary dynamics, i.e., low speciation and extinction rates and low degrees of faunal turnover, that have been demonstrated in studies of other marine invertebrate taxa (Sepkoski Jr, 1998; Stanley & Powell, 2003; Bonelli & Patzkowsky, 2011). However, Ramsbottom (1981), Kullmann (1985), Becker & Kullmann (1996), and Wiedmann & Kullmann (1996) did cogently argue that this was not the case for cephalopods. More recently, Balseiro (2016) did document the existence of some profound evolutionary turnover in bivalves and brachiopods over the course of this interval in regions closer to the ice sheets, such as present-day western Argentina. Furthermore, Segessenman & Kammer (2018) showed that advanced cladid crinoids do display elevated rates of evolution and turnover during this time interval (although three other subclasses of crinoids do show subdued evolutionary rates), and fusulinid foraminifera also fit the pattern shown in the advanced cladids (Groves & Lee, 2008; Groves & Yue, 2009; Segessenman & Kammer, 2018).

There have been a variety of hypotheses proposed for the postulated decelerated macroevolutionary dynamics (albeit not necessarily in cephalopods) of the LPIA. Some studies contend that this pattern is a result of environmental changes linked to glacial cycling while others point to tectonic activity (Stanley & Powell, 2003; Powell, 2005; Fielding, Frank & Isbell, 2008; DiMichele et al., 2009; Falcon-Lang & DiMichele, 2010; Bonelli & Patzkowsky, 2011; Cecil, DiMichele & Elrick, 2014; Segessenman & Kammer, 2018). To date, many of the more recent studies focusing on the macroevolutionary dynamics of the LPIA have concentrated on benthic marine invertebrates (e.g.,  Stanley & Powell, 2003; Powell, 2007; Bonelli & Patzkowsky, 2011; Balseiro, 2016; Segessenman & Kammer, 2018) as they are highly diverse and very abundant. However, it is valuable to also investigate evolutionary patterns in pelagic marine invertebrates as these are also diverse and abundant organisms in late Paleozoic marine ecosystems (Landman, Tanabe & Davis, 1996; Monnet, De Baets & Klug, 2011; Klug et al., 2015; Korn, Klug & Walton, 2015). In particular, given the significant role that geographic factors play in speciation (Mayr, 1942; Eldredge & Gould, 1972; Jablonski, 1986; Brooks & McLennan, 1991; Wiley & Lieberman, 2011; Jablonski & Hunt, 2015; Pie & Meyer, 2017), we might expect that pelagic organisms, because of their innately greater dispersal ability (at least as adults), might show different patterns relative to taxa that were benthic (Rojas et al., 2017; Yacobucci, 2017). This greater dispersal ability might allow pelagic organisms to more fully occupy potentially available habitats than benthic organisms, which could lead to larger geographic ranges and also less change in geographic ranges through time. (In addition, there are certain paleoecological constraints that reduce the dispersal potential of cephalopods, such as minimum water depth required for vertical migration, Ward & Westermann, 1985; Ritterbush et al., 2014; RT Becker, pers. comm., 2019). It also could potentially influence patterns of speciation and extinction by dampening opportunities for geographic isolation and creating larger effective population sizes. Further, sea-level fall is known to cause regular and repeated patterns of extinction in ammonoids (Kullmann, 1983; Kullmann, 1985; House, 1985; Hallam, 1987; Becker & Kullmann, 1996; Wiedmann & Kullmann, 1996; Kaiser et al., 2011; Zhang et al., in press; and RT Becker, pers. comm., 2019).

This study focuses on cephalopods from the Pennsylvanian-early Permian (Morrowan, Atokan, Desmoinesian, Missourian, Virgilian, and Wolfcampian) in the Midcontinent Sea of the United States as knowledge of their systematic affinities, geographic distribution and overall diversity is relatively well understood (Miller, Dunbar & Condra, 1933; Newell, 1936; Plummer & Scott, 1937; Miller & Youngquist, 1949; Nassichuk, 1975; Boardman II et al., 1994; Landman, Tanabe & Davis, 1996; Kröger, 2005; Klug et al., 2015; Korn, Klug & Walton, 2015), the stratigraphy of the region is well constrained (Heckel, 2008; Heckel, 2013), and there are extensive exposures of fossiliferous units in the region. Moreover, at this time the Midcontinent Sea was bordered by the Antler Orogeny to the north, the Ancestral Rocky Mountain Orogeny to the west/northwest and the Ouachita Mountain belt to the south/southeast (as well as various structural arches), such that it constituted a distinct biogeographic region for marine invertebrates (Wells et al., 2007; Nelson & Lucas, 2011; Joachimski & Lambert, 2015).

The Late Paleozoic Ice Age (LPIA) was the longest lived glacial period of the Phanerozoic and is relatively well understood due to numerous stratigraphic, sedimentologic, paleontologic, and isotopic studies (e.g., Mii, Grossman & Yancey, 1999; Isbell, 2003; Stanley & Powell, 2003; Raymond & Metz, 2004; Montañez, 2007; Powell, 2007; Tabor & Poulsen, 2007; Fielding, Frank & Isbell, 2008; Heckel, 2008; DiMichele et al., 2009; Bonelli & Patzkowsky, 2011; Montañez & Poulsen, 2013; Balseiro, 2016; Roark et al., 2017; Segessenman & Kammer, 2018). Glacial cycling in the North American midcontinent region has received much study (e.g.,  Isbell, 2003; Heckel, 2008; Heckel, 2013). Modern synthesis of the glacial history indicates that the Morrowan to early Desmoinesian represented a localized glacial period, the late Desmoinesian to early Virgilian represented a widespread interglacial period with minor glaciation, and the late Virgilian to early Wolfcampian represented the apex of widespread glaciation (Montañez & Poulsen, 2013). Modeling predicts that sea-level oscillations in the late Pennsylvanian were between 50–100 m depending upon the number and volume of melting ice sheets, and that water temperatures are estimated to have been between 4–7 °C cooler during glacial maxima than inter-glacial periods (Heckel, 1986; Isbell, 2003; Montañez, 2007; Tabor, 2007; Heckel, 2008; Cecil, DiMichele & Elrick, 2014). The sea-level and temperature changes were likely to have had an important influence on species distribution and geographic range size during this time (Waterhouse & Shi, 2010). Perhaps cephalopod taxa would be less influenced by glacial sea-level cycles than benthic taxa, as these cycles are also known to cause variation in seafloor ventilation, with concomitant dysoxia/anoxia that is more severe for benthic taxa (A Dunhill, pers. comm., 2018). By contrast, sea-level fall is known to have caused ammonoid extinctions and Paleozoic cephalopods were sensitive to water temperature (RT Becker, pers. comm., 2019).

Materials and Methods

Taxa considered, stratigraphic correlation, specimens examined, and georeferencing

Seventy-nine species belonging to 26 genera (13 nautiloids and 13 ammonoids) of cephalopods in the Pennsylvanian-Permian North American Midcontinent Sea were considered (Table S1). These represent abundant, well preserved, and taxonomically well understood species for which we were able to obtain type material and collections material of sufficient quality to enable taxonomic assignments on a breadth of material. Other species from the mid-continent of North America certainly exist and adding these to our analyses could change our results. However, at this time it was not possible to consider these via obtaining type and other material for them and pursuing the significant additional taxonomic work this would entail. Therefore, results are based on consideration of what is essentially a random selection of some of the (albeit well known) species in the region and this analysis is best viewed as an initial approach to considering paleobiogeographic dynamics in the region. Range reconstructions relied on the occurrence records of specimens derived from a comprehensive consideration of the entire taxonomic literature on the taxa studied. In particular, the following publications were utilized: Cox (1857), Swallow (1858), McChesney (1860), Meek & Worthen (1860), Meek & Worthen (1870), White & St. John (1867), White (1889), Hyatt (1891), Hyatt (1893), Keyes (1894), Miller (1892), Smith (1896), Smith (1903), Girty (1911), Girty (1915), Mather (1915) Böse (1919), Böse (1920), Miller (1930), Sayre (1930), Miller, Dunbar & Condra (1933), Miller & Cline (1934), Miller & Owen (1934), Miller & Owen (1937), Miller & Owen (1939), Foerste (1936), Miller & Thomas (1936), Newell (1936), Plummer & Scott (1937), Elias (1938a), Elias (1938b), Miller & Moore (1938), Smith (1938), Miller & Furnish (1940a), Miller & Furnish (1940b), Miller & Furnish (1957), Teichert (1940), Clifton (1942), Miller & Unklesbay (1942), Young (1942), Sturgeon (1946), Miller, Lane & Unklesbay (1947), Miller & Downs (1948); Miller & Downs (1950), Miller & Youngquist (1947), Miller & Youngquist (1949), Miller, Youngquist & Nielsen (1952), Kummel (1953), Kummel (1963), Ruzhentsev & Shimanskiy (1954), Unklesbay (1954), Arkell et al. (1957), Unklesbay & Palmer (1958), Hoare (1961), Furnish, Glenister & Hansman (1962), McCaleb (1963), Gordon (1964), Miller & Breed (1964), Teichert et al. (1964), Furnish & Glenister (1971), Ruzhentsev & Bogoslovskaya (1971), Nassichuk (1975), Sturgeon et al. (1982), Hewitt et al. (1989), Boardman II et al. (1994), Kues (1995), White & Skorina (1999), Kröger & Mapes (2005), Furnish et al. (2009), and Niko & Mapes (2009) as well as from examination of all specimens, including types, housed in: the Division of Invertebrate Paleontology, Biodiversity Institute, University of Kansas (KUMIP); the University of Iowa Paleontology Repository (UI); and the Yale University Peabody Museum of Natural History (YPM). These institutions are among the most complete repositories of cephalopod diversity from this region and time and contain many of the type specimens of the species examined. Moreover, all specimens used in the analysis were personally examined and taxonomically-vetted via consideration of the literature, relevant type specimens, and other material, with species assignments and determinations made by the first author. Over 1,100 specimens were identified to species level in this study (Kolis, 2017). We chose to focus on the particular species considered, rather than downloading data from the Paleobiology Data Base (PBDB), as we wanted to be able to personally validate the taxonomic identity of specimens using collections data in conjunction with the literature in order to present more rigorously corroborated hypotheses about the geographic distributions of species. We consider this approach to be complementary to those approaches that utilize the PBDB in paleobiogeographic studies. On the one hand, our approach did limit the number of species we were able to consider. On the other hand, we believe it is quite important to evaluate hypotheses about systematic affinities of fossil specimens, the actual data of the fossil record themselves, in detail and thereby accurately define the taxonomic units considered. Given that species represent key macroevolutionary units in nature (Eldredge, 1989; Wiley & Lieberman, 2011; Hendricks et al., 2014), correctly characterizing them taxonomically, and thus validating the scope of their geographic distributions, is critical. Moreover, it has recently been shown by Marshall et al. (2018) that incorporating museum specimen data in the manner that our study has can greatly expand, enhance, and improve knowledge of geographic distributions of fossil species, relative to studies that only utilize data from the PBDB. In the case of some species, ∼30% of the total considered, our analyses indicated moderate changes in stratigraphic range (addition of a stage, etc.) relative to what is presented in the PBDB. This happened primarily because via this study we were able to identify specimens to species that previously had been treated as indeterminate at the species level, or we were able to determine that specimens had previously been mis-identified to species.

Specimens were assigned to the Virgilian, Missourian, Desmoinesian, Atokan, Morrowan, or Wolfcampian stages using the USGS National Geologic Map Database (USG, 2017; Sawin et al., 2006; Sawin et al., 2008; Sawin et al., 2009; Zeller, 1968; Pope, 2012; Heckel, 2013). The temporal boundaries of stages were derived from Davydov, Korn & Schmitz (2012) (Table S2). It is important to note that the boundaries of international stages are based on few geochronological tie points and the correlation of the North American stage boundaries with these is arbitrary; also, some of the boundaries used are still being researched (RT Becker, pers. comm., 2019.) In addition, while more resolved stratigraphic assignment to biostratigraphic zone is possible for units in Europe (e.g.,  Davydov & Leven, 2003), the northern Appalachian Basin of North America (e.g.,  Heckel, Barrick & Rosscoe, 2011), and parts of the North American midcontinent (e.g., Boardman II et al., 1994; Heckel, Barrick & Rosscoe, 2011), it is less tractable to associate the boundaries of the biostratigraphic zones from the North American midcontinent with radiometric dates for the stratigraphic units and regions considered herein. Furthermore, the museum specimens considered herein lacked the information needed to make it possible to constrain them to biostratigraphic zone, only stage. For this reason, it was unfortunately not possible to consider changes in geographic range, nor rates of speciation and extinction, at a temporal scale more resolved than stage. Although this is often the standard degree of temporal resolution used in a variety of paleobiogeographic studies, it does entail that we were not able to discern events transpiring more rapidly than the time scale of stage. This means that we will be missing important patterns; although speciation and extinction does not appear to frequently be transpiring within stage boundaries in this region, at least sometimes it is, and moreover geographic range shifts by species were certainly happening within these boundaries.

All specimen localities were georeferenced during the course of the study. GEOLocate (Rios & Bart Jr, 2018) and the MaNIS Georeferencing Calculator (Wieczorek & Wieczorek, 2015) were used to obtain coordinates and uncertainty radii. All points were calculated in decimal degrees within the WGS84 model in the GEOLocate (Rios & Bart Jr, 2018) world topo layer to ensure consistency and accuracy in determinations. Most uncertainty radii were less than 10 kms. Any specimens with questionable locality information were excluded from analyses, as were specimens with an uncertainty radius larger than the county they were contained within. This left 950 specimens (Table S1) to use in range reconstruction and statistical analysis of geographic range through geologic time. All statistical analyses were performed using Minitab® Statistical Software Minitab v. 17 (Minitab, 2016) and RStudio (2017).

Range reconstruction using GIS

Methods for range reconstruction follow Rode & Lieberman (2004), Rode & Lieberman (2005), Stigall & Lieberman (2006), Hendricks, Lieberman & Stigall (2008), Myers & Lieberman (2011), Myers, MacKenzie & Lieberman (2013), and Dunhill & Wills (2015). In particular, after specimen occurrence data were georeferenced and assigned to temporal bins, Excel CSV files were compiled for the occurrence points for all specimens within species. CSV files were imported into ArcGIS v. 10.3 (Esri, Redlands, CA, USA) and layers were created using geographic coordinate system ‘WGS 1984” and projected coordinate system ‘WGS 1984 World Mercator’ (Fig. 1). These layers were input into PaleoWeb (The Rothwell Group LP, 2016) to rotate coordinates into continental configuration and geographic position of the midcontinent region during the Pennsylvanian-early Permian (Fig. 2). These paleo-coordinate layers were then re-projected into ArcMap (Esri, Redlands, CA, USA).

Figure 1 Distribution of Pennsylvanian and early Permian cephalopods.

(A) Distribution of Pennsylvanian nautiloid and ammonoid data points (red) and (B) early Permian nautiloid and ammonoid data points (blue) across the midcontinent region of North America. Plotted using ArcGIS v. 10.3 (Esri, Redlands, CA, USA) software at 1: 20,000,000.

Figure 2 Occurrence points of Metacoceras sp. and Mooreoceras sp.

For the Virgilian, shown on possible paleogeography of that stage, at 1:1,000,000,000 scale; plotted using PaleoWeb (Rothwell Group LP, 2016).

Geographic range values were calculated for each species (Table S3) using minimum bounding geometry. This method has been shown to provide the most accurate procedure for reconstructing changes in geographic range, especially for fossil taxa (Darroch & Saupe, 2018). Convex hulls or buffers were given to every specimen occurrence point in each species and these shapefiles were re-projected in ‘South America-Albers Equal Area Conic’. This model was used to accommodate the rotation of species occurrence coordinates into the southern hemisphere during the late Paleozoic. Species with three or more occurrence points were given a convex hull that spanned the entire area between occurrences (see Rode & Lieberman, 2004; Hendricks, Lieberman & Stigall, 2008; Myers & Lieberman, 2011; Darroch & Saupe, 2018). In this way, multiple occurrence points were combined to recreate the geographic range of a single species. Species with only one occurrence point were given a 10 km2 buffer; species with just two occurrence points were given a 10 km2 wide buffer which was used, in conjunction with their distance, to derive an area value (following Rode & Lieberman, 2004; Rode & Lieberman, 2005; Hendricks, Lieberman & Stigall, 2008; Myers & Lieberman, 2011; Myers, MacKenzie & Lieberman, 2013). Species geographic range size data were tested for normality within each temporal stage using the Anderson-Darling normality test (this is a commonly used test to assess normality, see Sokal & Rohlf, 1994).

Assessing fossil record bias

A common concern when studying the fossil record is that there might be biases that could lead to inaccurate or artifactual findings. This concern can be manifold, but the two most pertinent issues here involve incomplete sampling and/or issues of stratigraphic bias. While it is important to be aware of the fact that the fossil record is incomplete, it is worth recognizing that there is a large body of research that demonstrates that many of the biogeographic patterns preserved in the fossil record, particularly in marine settings, represent real biological phenomena, rather than taphonomic artifacts (Myers & Lieberman, 2011; Rook, Heim & Marcot, 2013; Dunhill & Wills, 2015), although that does not mean that such artifacts played no role in this study. Further, it is also prudent to realize that sampling bias is a common issue in studies of extant biodiversity and species distribution, and much work needs to be done in this area to alleviate the biases of the extant biota (Lieberman, 2002; Carrasco, 2013).

The possibility that biases in the fossil record might lead to artifactual results was assessed in a few different ways. First, the relationship between outcrop availability and the geographic range of Pennsylvanian and Permian cephalopods was determined (see Myers & Lieberman, 2011). A percent coverage table of the range size of species overlaid against temporal outcrop availability was created using ArcGIS v. 10.3 (Esri, Redlands, CA, USA). A low percentage of overlap between range size and outcrop area would suggest species distributions are more likely to reflect ‘real’ biogeographic patterns while a high percentage of overlap would suggest the presence or absence of outcrop was significantly influencing results (Myers & Lieberman, 2011; Myers, MacKenzie & Lieberman, 2013) (however, see also Dunhill, 2012 for an alternative viewpoint). The second test used was an “n-1” jackknifing analysis (see Myers & Lieberman, 2011; Myers, MacKenzie & Lieberman, 2013). This procedure sub-sampled species range size within each temporal bin to test the resilience of data to outliers. Mean range size estimations were generated for each temporal bin; these were input into a one-way ANOVA to compare jackknife estimates with the initial geographic range size estimates (Myers & Lieberman, 2011; Myers, MacKenzie & Lieberman, 2013). Finally, a Pearson rank correlation test was performed to test the association of occurrence points and geographic range size; a close correlation would indicate that reconstructed ranges were very much dependent on sampling and suggest that reconstructed biogeographic patterns might be an artifact of a biased fossil record (Myers, MacKenzie & Lieberman, 2013).

Speciation and extinction rate calculations

Speciation and extinction rates were calculated in order to consider macroevolutionary dynamics in cephalopods from the Late Paleozoic Midcontinent Sea. Macroevolutionary rates were calculated using the following equation, presented in Foote (2000) and Rode & Lieberman (2005): Nf=N0ert

where N0 is the species richness at the beginning of a temporal bin, Nf is the species richness at the end of a temporal bin, t is the duration of a temporal bin, and r is the total rate of diversity change. The temporal bins used were North American stages (Table S2). Species richness values (Nf) were determined for each temporal bin and were parsed into ‘carry-over’ (N0) and ‘new’ species richness values to ensure the accuracy of speciation and extinction rate calculation. In this way, it was possible to calculate the rate of diversity change between bins. For example, rAtokan = (ln N0−Desmoinesian − ln N0−Atokan)/tAtokan. Speciation rate within each temporal bin was calculated using the equation SAtokan = (ln Nf−Atokan − ln N0−Atokan)/tAtokan, and extinction rate within each temporal bin was calculated using the equation EAtokan = SAtokan − rAtokan for each temporal stage (Foote, 2000; Rode & Lieberman, 2005).

Results

Paleobiogeographic patterns

Geographic range data were analyzed separately across all cephalopods and individually for both nautiloids and ammonoids. As mentioned above, species geographic range size data were tested for normality within each temporal stage using the Anderson-Darling normality test (see Sokal & Rohlf, 1994). Range size data within each temporal stage were not normally distributed for any data combination (P < 0.005). Instead, distributions were left skewed across all temporal stages for every data grouping. Data were subsequently log-transformed to normalize data, and statistical analyses were performed on both original and transformed data.

In general, geographic range size (either mean of transformed data or median of original) of ammonoids and nautiloids increases during the Missourian and Virgilian stages (Fig. 3), which was a time of sea-level rise due to warming during an interglacial (Isbell, 2003; Montañez & Poulsen, 2013), such that there may be an association between the sea-level rise and the increase in geographic range. Another possibility is that there was some change in taphonomic or collecting conditions associated with Virgilian strata that made it easier to discern the actual biogeographic distributions of species at this time, relative to other time intervals (G Piñeiro, pers. comm., 2018). However, none of the changes in geographic range were statistically significant, so it is not possible to infer strong correlation between the sea-level rise, or possible taphonomic factors, and the range expansion. For instance, Mann–Whitney U tests, a non-parametric test used to compare two sample medians (see Sokal & Rohlf, 1994), found no statistically significant changes (at P ≤ 0.05) in median geographic range size for any temporal stages separately across all studied cephalopods, as well as individually for nautiloids and ammonoids, even prior to correction for multiple comparisons. This is because for the cephalopods studied median range values are constant through time (79 km2). Mean values (which are also relevant for understanding patterns of change in the data, G Piñeiro, pers. comm., 2019) do show more change through time in our data than the corresponding median values, as might be expected, but median values are better to focus on for statistical purposes when the data are not normally distributed, as is the case herein.

Figure 3 Mean geographic range size in km2 of cephalopods through time.

Nautiloid species (A) and ammonoid species (B) range changes occur but are not statistically significant when analyzed using non-parametric tests (note, median range size data not graphed but for all cephalopods they are 79 km2 for all time intervals, for ammonoids they are 78.5 km2 for the Desmoinesian and Wolfcampian and 79 km2 for all other time intervals, and for nautiloids they are 79 km2 for all time intervals) or when log transformed data are analyzed using parametric tests (note log transformed data not graphed but mean transformed values for all cephalopods are 5.51 [standard error 0.75] for the Morrowan, 4.05 [standard error 1.02] for the Atokan, 4.36 [standard error 0.49] for the Desmoinesian, 5.65 [standard error 0.49] for the Missourian, 5.96 [standard error 0.79] for the Virgilian, and 4.31 [standard error 0.52] for the Wolfcampian).

The same was true for two-sample t-tests (see Sokal & Rohlf, 1994) performed on log-transformed data which again found no statistically significant changes (at P ≤ 0.05) in mean geographic range size though time, even prior to employing a statistical correction needed in the case when there are multiple comparisons. Again, recall that mean range size data are shown in Fig. 3, and the differences among log-transformed data through time are far less substantial (and ultimately not significant). Furthermore, a one-way ANOVA, either with or without the assumption of equal variance, failed to find any significant differences (at P ≤ 0.05) between stages for log-transformed mean geographic range size across all cephalopods as well as individually for nautiloids and ammonoids. Still, it is worth noting that changes in range size are occurring through time, most notably in the Virgilian, and these could be related to climatic changes that occurred then, and also changes in the paleogeography of the region, although in the absence of statistical evidence we could not convincingly document such a link in the present study. However, it is important to note that previous studies (e.g., Ramsbottom, 1981) have documented such a link.

Analysis of macroevolutionary rates

Speciation rate (S) and extinction rate (E) were calculated for the Atokan, Desmoinesian, Missourian, and Virgilian stages across all selected cephalopods and within selected nautiloids and ammonoids, respectively. The S and E presented across all selected cephalopods are comprised of two calculations; one calculation included taxa that only occurred in a single temporal stage (singletons) (Table 1; Fig. 4), while the other calculation excluded taxa that occurred in a single temporal stage (Table S4). S and E were also calculated for ammonoids and for nautiloids including (Tables S5 and S6) and excluding taxa that occurred in a single stage (Tables S7 and S8). Note, due to the dependence of calculations on diversity metrics from both adjacent stages, it is not possible to accurately calculate the rate of biodiversity change (R), or S and E for the first stage considered, the Morrowan, nor R or E for the last stage considered, the Wolfcampian (these are thus left blank in Table 1 and Tables S4–S8). While it might have been possible to infer S and E using other methods, to do so would exaggerate the significance of edge effects and thus be problematic (Foote, 2000). A problem with including singleton taxa is that since they speciate and go extinct in the same interval there will always be a direct one to one correlation between S and E (Vrba, 1987; Foote, 2000). This is why for studies considering the relationship between S and E it is recommended that singletons be excluded (Vrba, 1987; Foote, 2000). However, when singletons are not included, a higher proportion of ammonoids cannot be considered, as many of these have short biostratigraphic ranges (RT Becker, pers. comm., 2019). To address each of these concerns we have presented calculations both with and without singletons.

Table 1 Speciation rates (S) per millions of years (Myr), extinction rates (E) per Myr, and rate of turnover (R) per Myr, for each stage across all cephalopods, with species richness values, species carryover from the previous stage, new species originating in the stage, No (the initial number of species), Nf (the final number of species), and duration (in Myr) also given.

Stage	Species richness	Species carryover	New species	No	Nf	Duration	R	S	E	
Wolfcampian	13	7	6	7	13	14		0.0442		
Virgilian	38	32	6	32	38	5	−0.3040	0.0343	0.3383	
Missourian	55	33	22	33	55	3	−0.0103	0.1703	0.1805	
Desmoinesian	41	12	29	12	41	3	0.3372	0.4096	0.0724	
Atokan	15	7	8	7	15	2	0.2694	0.3811	0.1116	
Morrowan	8	0	8	0	8	6				

Figure 4 Speciation and extinction rates through time.

Values given in per Myr and derived from Table 1.

Across all cephalopods studied, S was high in the Atokan and Desmoinesian, fell in the Missourian, and reached very low levels in the Virgilian and Wolfcampian (Fig. 4). By contrast, E was low in the Atokan and Desmoinesian, began to rise in the Missourian, and reached even higher levels in the Virgilian (Fig. 4). Essentially, across all cephalopods examined, when S is high, E is low, and when S is low, E is high. This is potentially contrary to the pattern expected with an ecological opportunity model of speciation (Simões et al., 2016), although the specific processes driving the diversification could not be determined at this time. However, it is possible that when S was high there may have been many short-lived species that could not be sampled that were actually going extinct, and this phenomenon would artificially depress E. To consider this in more detail, what is truly needed is a zone by zone analysis of all cephalopod species known from the North American midcontinent (RT Becker, pers. comm., 2019).

As expected, S and E are lower when singletons are excluded (see Table 1, S4). Segessenman & Kammer (2018) found in their macroevolutionary study on crinoids from this interval that including or excluding singletons substantially influenced their results, but in our study including or excluding these did not produce a substantial change. Notably, S and E patterns diverge somewhat between ammonoids and nautiloids when considered individually (and the patterns in nautiloids better match the overall patterns across all the cephalopods studied). For instance, in nautiloids S is high in the Atokan and Desmoinesian, then declines to moderate in the Missourian, and is at its lowest in the Virgilian and Wolfcampian (Table S6), whereas in ammonoids S is only high in the Atokan, declines to moderate in the Desmoinesian, declines somewhat more in the Missourian and then remains essentially constant through the Wolfcampian (Table S5). In addition, E is low in ammonoids during the Desmoinesian and Missourian but high in the Atokan and Virgilian (Table S5), whereas in nautiloids there are no observed extinctions during the Atokan; values remain quite low for nautiloids in the Desmoinesian, rise somewhat in the Missourian, and then rise again in the Virgilian (Table S6).

An important caveat regarding the calculation of S is that many of the species analyzed belong to genera that were widely distributed beyond the Midcontinent Sea during the late Paleozoic. Thus, although none of the species considered in these analyses occurred outside of the Midcontinent Sea, their close relatives did. It is conceivable that while speciation events and rates by necessity are herein treated as occurring in situ, this might not always have been the case. Instead, some speciation events could have occurred outside of the Midcontinent Sea with subsequent invasion events into that region. These invasions would appear as in situ speciation events in this analysis, although they actually were not. In the absence of phylogenetic hypotheses for the genera considered it is not currently possible to consider how much of the pattern pertaining to speciation rate shown in Fig. 4 is due to invasion instead of speciation such that both might be playing a role (Metacoceras is one example where the genus occurs well outside of the North American mid-continent, it is known to occur in beds ∼100 kms southeast of Moscow, Russia, such that some of the cladogenetic events involving this genus might comprise instances of invasion). Further, a related phenomenon could affect the calculation of E: at times what were treated as extinction events might have simply been local extinctions in the Midcontinent Sea which could have included emigration to other regions. As mentioned previously, it does not appear that any of the species considered occur outside of the Midcontinent Sea, but a phylogenetic hypothesis for these groups would be valuable for considering this issue in greater detail.

Relationship between biogeography and macroevolutionary rates

Across all the cephalopods studied, mean geographic range size increased during the Virgilian (and in ammonoids first in the Missourian but then more prominently in the Virgilian) and declined in the Wolfcampian (Fig. 3); speciation rates were generally high in the Atokan and Desmoinesian and fell in the Virgilian (Fig. 4); extinction rates were generally low in the Atokan and Desmoinesian and rose in the Virgilian (Fig. 4). The Pearson correlation test in Minitab 17 (Minitab, 2016) was used to examine the association between geographic range and either speciation rate extinction rate in greater detail. No significant (at P ≤ 0.05) correlation between speciation or extinction rate and range size was found across all cephalopods or within ammonoids or nautiloids individually (Table 2). However, in cases the values approach P = 0.05. For instance, the association between decreasing geographic range size and increasing extinction for all cephalopods and for ammonoids alone, so it is clear that generally there is some association between the two, but unfortunately significant support at the .05 level is lacking. We note that numerous previous studies have documented an association between decreasing geographic range size and increasing extinction rate (e.g., Vrba, 1980; Jablonski, 1986; Eldredge, 1989; Stanley, 1990; Jablonski & Roy, 2003; Rode & Lieberman, 2004; Rode & Lieberman, 2005; Kiessling & Aberhan, 2007; Payne & Finnegan, 2007; Stigall, 2010; Dunhill & Wills, 2015; Jablonski & Hunt, 2015; Orzechowski et al., 2015; Saupe et al., 2015; Castiglione et al., 2017; Pie & Meyer, 2017; Lam, Stigall & Matzke, 2018; Schneider, 2018) and thus this a very robust phenomenon in general and likely to be operating to some extent herein. However, over this time interval and for this particular group of species the association is not statistically significant (Table 2), probably because sample sizes are not large, and further this is likely because many taxa were culled by the late Mississippian extinction (M Powell, pers. comm., 2018). Further, sample size could also be influencing the results pertaining to changes in geographic range size through time (G Piñeiro, pers. comm., 2019).

Table 2 Pearson correlation test for association between S and geographic range and E and geographic range across all cephalopods and for ammonoids and nautiloids individually, with Pearson’s r and P-values given.

Taxon–speciation	Pearson’s r	P-value	Taxon–extinction	Pearson’s r	P-value	
All Cephalopods-S	−0.541	0.347	All Cephalopods-E	0.925	0.075	
Nautiloids-S	−0.463	0.432	Nautiloids-E	0.913	0.087	
Ammonoids-S	−0.519	0.370	Ammonoids-E	0.803	0.197	

Analysis of fossil record bias

The low percentage of overlap between cephalopod species geographic ranges and the availability of outcrop, less than 1% in 29 out of 30 species (Table S9; the one species with a larger percentage value, “Orthoceras” kansasense, occurs throughout the Midcontinent Sea), suggests the results are not simply an artifact of an incomplete fossil record, at least pertaining to outcrop availability or changes in the paleogeography of the region. The “n-1” jackknifing analysis also supports the robustness of the reconstructed ranges, as no statistically significant differences were found between the mean of the reconstructed and subsampled range values for any time interval (all P-values > 0.9), suggesting that one or a few occurrence records are not having a major influence on biogeographic patterns. Similar results were found in other taxa and time periods by Hunt, Roy & Jablonski (2005), Myers & Lieberman (2011), and Myers, MacKenzie & Lieberman (2013), although Dunhill, Hannisdal & Benton (2014) did find some association between outcrop area and diversity in the case of the marine fossil record of Great Britain. Finally, the Pearson correlation test shows no correlation (−0.055, P-Value = 0.789) between the number of occurrence points and geographic range size; this provides further evidence that the biogeographic signatures of late Paleozoic cephalopods are unlikely to be simply an artifact of the fossil record.

Diversity patterns

Across all cephalopods, species richness increased from the Morrowan to the Atokan, peaked in the Desmoinesian, and decreased through the Wolfcampian (Fig. S1). A similar pattern is seen in the nautiloids (Fig. S2). However, the ammonoids (Fig. S3) demonstrate an earlier peak in the Atokan, followed by a Desmoinesian to Virgilian plateau, with a decrease in the Wolfcampian. This indicates that the data from nautiloids are most influencing the recovered patterns (G Piñeiro, pers. comm., 2019). Notably, previous studies of late Paleozoic brachiopod communities in Bolivia showed a consistent trend between diversity and glacial cycling with increased diversity during glacial periods and decreased diversity during inter-glacial periods (Badyrka, Clapham & Lopez, 2013). However, there seems to be less consistency between species richness trends and glacial cycling in the Midcontinent Sea. For instance, there is an increase in cephalopod species richness throughout the Morrowan to Desmoinesian associated with localized glaciation, and an interglacial period with generally minor glaciation is associated with a decrease in cephalopod species richness from the Desmoinesian to Virgilian, yet by contrast widespread glaciation is associated with a decrease in species richness from the Virgilian to the Wolfcampian. Important points, however, are that these are raw diversity patterns, and sample standardized diversity patterns show a different result (M Powell, pers. comm., 2018), and further that brachiopods and cephalopods can show different behaviors in response to climatic changes (G Piñeiro, pers. comm., 2019).

Discussion

Geographic range shifts through time are one of the pervasive phenomena in the history of life; these are manifest both within species and higher-level clades, occur at a number of different time scales, and are frequently linked to climatic change (Wiley & Lieberman, 2011). Specific examples do come from the late Paleozoic, a time of extensive climate change including profound glaciation along with numerous glacial and interglacial cycles and associated cycles of sea-level rise and fall (Montañez & Poulsen, 2013). (Previous studies of ammonoids have shown that these changes in sea-level may have caused more significant changes in biogeographic ranges of taxa than temperature changes during this time period, and other time periods as well (Hallam, 1987; Hartenfels & Becker, 2016; Zhang et al., in press). Those changes impacted patterns of geographic range in both terrestrial plant (e.g., DiMichele et al., 2009; Falcon-Lang & DiMichele, 2010) and marine invertebrate ecosystems (e.g., Ramsbottom, 1981; Leighton, 2005; Powell, 2007; Waterhouse & Shi, 2010; Balseiro & Halpern, 2019). When it comes to marine invertebrates from this time interval, most of the focus has been on the highly diverse benthic faunas (e.g., Stanley & Powell, 2003; Powell, 2007; Bonelli & Patzkowsky, 2011; Balseiro, 2016; Segessenman & Kammer, 2018; Balseiro & Halpern, 2019); however, taxa that have a pelagic life style are also worth examining. Herein, 79 pelagic species of cephalopods were examined for patterns of range size change using GIS and although in general these species exhibit some evidence for changes in geographic range size (Fig. 3) especially in the Virgilian, and to a lesser extent in the Missourian, those changes were not statistically significant, making it hard to directly tie them to climate changes. However, there is strong evidence that climate change played a prominent role in influencing geographic range of cephalopods from other regions during this time period (e.g., Ramsbottom, 1981) and indeed in cephalopods from other time periods (e.g., Hallam, 1987; Jacobs, Landman & Chamberlain Jr, 1994; Kaiser et al., 2011; Hartenfels & Becker, 2016; Zhang et al., in press). In a similar vein, many paleontological studies have demonstrated that species with larger geographic ranges tend to have lower extinction rates than species with narrower geographic range sizes (e.g., Vrba, 1980; Jablonski, 1986; Eldredge, 1989; Stanley, 1990; Rode & Lieberman, 2004; Stigall & Lieberman, 2006; Payne & Finnegan, 2007; Stigall, 2010; Hopkins, 2011; Dunhill & Wills, 2015). Again, this phenomenon is not found to be statistically significant in the case of the late Paleozoic cephalopod species considered herein (Table 2), but there is some general quantitative evidence for the phenomenon.

There may be a few different explanations for these findings. First, it may be that some cephalopod species were not significantly affected by the glacial-interglacial climatic cycles transpiring within the Late Paleozoic Midcontinent Sea. A second possible explanation, perhaps coupled to the first, is that since cephalopods are highly mobile relative to benthic marine invertebrates such as gastropods, bivalves, brachiopods, etc., they can more easily occupy a greater portion of their potential range. Further, perhaps the available potential range of cephalopod species does not change much in glacial relative to interglacial regimes. This may seem unlikely given the vast fluctuations in sea level occurring at the time, but pelagic marine organisms, because of their ease of dispersal, may more easily maintain consistent geographic ranges relative to benthic counterparts. Another possible explanation for the pattern retrieved is that, given the limits of stratigraphic correlation, sample size, and the completeness of the fossil record, it was necessary for the analyses of species distribution conducted herein to focus on the time scale of geological stages, whereas in actuality there were climatic changes occurring within stages (Heckel, 2008; Heckel, 2013); these certainly did cause fluctuations in species’ geographic ranges within stages, but simply could not be observed in the present study. The inability to observe changes in geographic range size of species at a scale more resolved than stage, in particular, likely played an important limiting role in the conclusions that could be derived. For instance, other studies such as Ramsbottom (1981) have looked at European taxa from the same time period, but focused at the level of zones, and did find a strong association between climate, sea-level, and geographic distribution. A final set of explanations are related to the issue of sampling. For instance, it was more difficult for the analyses presented herein to detect a relationship between geographic range size and macroevolutionary rate because speciation and extinction rates could only be calculated for four stages. Although we did not observe a substantial amount of speciation and extinction occurring within stage boundaries, certainly being able to consider more stages would have enhanced our ability to retrieve patterns. We suspect that another important explanation for our results is the relatively limited number of species that could be considered herein. An expansion in the number of taxa considered could absolutely change our results in various ways, including via increasing statistical power. Thus, what is presented herein should only be treated as preliminary results that require further data and additional testing. We would note, though, that detailed taxonomic vetting of specimens, including through comparison of type material, especially involving taxonomic studies conducted in some cases more than 70 years ago, requires a significant amount of time investment. Thus, dramatic expansions to this dataset would require concomitant investments of time. However, other datasets such as AMMON and the PBDB could be used if one did not feel it was necessary to spend time vetting taxonomic assignments. Although we posit that it is important to vet taxonomic assignments that may be outdated, we would assert that our approach should be viewed as complementary to approaches that rely on mining currently existing paleontologically oriented databases, and that both types of approaches have value. As a final possible explanation for our results, we further note that a common concern when studying the fossil record is the potential role biases can play. This concern can be manifold. It is somewhat obviated by the results presented herein regarding the apparent quality of the fossil record, but that does not mean that there are no inherent problems with the cephalopod record that are at present difficult to ascertain; these could be influencing the results retrieved in some at present unspecified way.

There is, however, another finding contrary to what might typically be expected for the late Paleozoic that is worth mentioning. That is the fact that there seems to have been at least some moderate degree of evolutionary diversification and turnover within cephalopods, such that species diversity did fluctuate throughout the Pennsylvanian and early Permian. Pennsylvanian rates of macroevolution are typically classified as ‘sluggish’ or ‘stolid’ across all marine animals, and Sepkoski Jr (1998) formalized the notion that there was a marked decline in evolutionary rates of Carboniferous and Permian marine faunas. Stanley & Powell (2003) reiterated this result and identified low mean macroevolutionary rates for marine invertebrate taxa. Bonelli & Patzkowsky (2011) also documented a pattern of low turnover in the face of major episodes of sea-level rise and fall due to climatic change. The results from the analyses presented herein could indicate that macroevolutionary rate in the case of late Paleozoic cephalopods was more dynamic than often thought, supporting the conclusions of a variety of other important studies considering late Paleozoic ammonoid diversity including Kullmann (1983), Kullmann (1985), House (1985), Becker & Kullmann (1996), Wiedmann & Kullmann (1996), and Kullmann, Wagner & Winkler Prins (2007). One possible reason why cephalopods may show a higher rate of diversification than other groups is that they were a fairly evolutionarily volatile group (Lieberman & Melott, 2013); thus, relative to many other marine invertebrate groups, they had relatively high rates of speciation and extinction (Stanley, 1979; Jacobs, Landman & Chamberlain Jr, 1994; Landman, Tanabe & Davis, 1996; Monnet, De Baets & Klug, 2011; Klug et al., 2015; Korn, Klug & Walton, 2015). However, this may not be the entire explanation, as some other groups also show elevated rates of speciation and extinction during this time interval. For instance, Balseiro (2016) and Balseiro & Halpern (2019) did document evolutionary turnover at high latitudes, and elevated evolutionary rates have also been found in fusulinid foraminifera (Groves & Lee, 2008; Groves & Yue, 2009) and advanced cladid crinoids (Segessenman & Kammer, 2018). Ultimately, we support the contention raised by Segessenman & Kammer (2018) that patterns from a few individual groups do not refute the general pattern of sluggish macroevolution postulated for this time period in the history of life. The results may lend credence to the notion that macroevolutionary patterns across all marine animals are rarely unitary for any one time period in the history of life, and instead often tend to be variegated.

Conclusions

Patterns of range size change in late Paleozoic cephalopods from the North American Midcontinent Sea were investigated using GIS. These species do exhibit some evidence for changes in geographic range size through time, but the changes were not statistically significant nor could they be directly tied to climate change. Further, in contradistinction to what is usually found in the fossil record, cephalopod species with larger geographic ranges were not found to have lower extinction rates than species with narrower geographic ranges. These distinctive patterns may perhaps be related to the fact that cephalopods are pelagic and highly mobile, at least relative to many benthic marine invertebrates, but it may also be due to the fact that only 79 species could be considered in our study, or to the fact that we were constrained to analyze patterns at the temporal level of stage. Finally, the group shows more evolutionary diversification and turnover during the Pennsylvanian and early Permian than is typical of other marine invertebrate groups and this could be related to the fact that cephalopods are an evolutionarily volatile group.

Supplemental Information

Figure S1 Late Paleozoic cephalopod species richness through time in the Midcontinent Sea

Time represented by North American temporal stage.

Click here for additional data file.

Figure S2 Late Paleozoic nautiloid species richness through time in the Midcontinent Sea

Time represented by North American temporal stage.

Click here for additional data file.

Figure S3 Late Paleozoic ammonoid species richness through time in the Midcontinent Sea

Time represented by North American temporal stage.

Click here for additional data file.

Table S1 List of cephalopod species considered and museum specimens used in range reconstructions, arranged alphabetically by genus and then species for Ammonoidea and Nautiloidea

Click here for additional data file.

Table S2 Temporal boundaries (from Davydov, Korn & Schmitz, 2012) used for calculations of Late Paleozoic cephalopod speciation and extinction rates

Click here for additional data file.

Table S3 Geographic range values through time in km2 by stage (youngest to oldest from left to right) for the species considered in the analysis

”0” indicates species is absent.

Click here for additional data file.

Table S4 Speciation rates (S) per Myr, extinction rates (E) per Myr, and rate of turnover (R) per Myr, for each stage across all cephalopods, with species that occur in a single stage excluded

Species richness values, species carryover from the previous stage, new species originating in the stage, No (the initial number of species), Nf (the final number of species), and duration (in Myr) also given.

Click here for additional data file.

Table S5 Speciation rates (S) per Myr, extinction rates (E) per Myr, and rate of turnover (R) per Myr, for each stage for ammonoids, with species that occur in a single stage included

Species richness values, species carryover from the previous stage, new species originating in the stage, No (the initial number of species), Nf (the final number of species), and duration (in Myr) also given.

Click here for additional data file.

Table S6 Speciation rates (per Myr), extinction rates (per Myr), and rate of turnover (R) per Myr, for each stage for nautiloids, with species that occur in a single stage included

Species richness values, species carryover from the previous stage, new species originating in the stage, No (the initial number of species), Nf (the final number of species), and duration (in Myr) also given.

Click here for additional data file.

Table S7 Speciation rates (S) per Myr, extinction rates (E) per Myr, and rate of turnover (R) per Myr, for each stage for ammonoids, with species that occur in a single stage excluded

Species richness values, species carryover from the previous stage, new species originating in the stage, No (the initial number of species), Nf (the final number of species), and duration (in Myr) also given.

Click here for additional data file.

Table S8 Speciation rates (S) per Myr, extinction rates (E) per Myr, and rate of turnover (R) per Myr, for each stage for nautiloids, with species that occur in a single stage excluded

Species richness values, species carryover from the previous stage, new species originating in the stage, No (the initial number of species), Nf (the final number of species), and duration (in Myr) also given.

Click here for additional data file.

Table S9 Percent coverage table of the range size of various species compared with available outcrop, by North American stage

Click here for additional data file.

Thanks to Chris Beard, Kirsten Jensen, Julien Kimmig, and Luke Strotz for very helpful discussions on this work and thanks to them and Matthew Powell, Alexander Dunhill, Ralph Thomas Becker, Dieter Korn, Graciela Piñeiro, Thomas Algeo, and Wolfgang Kiessling for comments on previous versions of the manuscript. Thanks to Julien Kimmig for assistance with collections related matters and for providing access to specimens in the KUMIP; thanks to Tiffany Adrain for assistance with collections related matters and for providing access to specimens in the UI; and thanks to Susan Butts for assistance with collections related matters and for providing access to specimens in the YPM. Thanks to Michelle Casey and Erin Saupe for assistance with stratigraphic correlations and use of GIS.

Additional Information and Declarations

Competing Interests

Author Contributions

Data Availability

Bruce S. Lieberman is an Academic Editor for PeerJ.

Kayla M Kolis conceived and designed the experiments, performed the experiments, analyzed the data, contributed reagents/materials/analysis tools, prepared figures and/or tables, authored or reviewed drafts of the paper, approved the final draft.

Bruce S Lieberman conceived and designed the experiments, contributed reagents/materials/analysis tools, prepared figures and/or tables, authored or reviewed drafts of the paper, approved the final draft.

The following information was supplied regarding data availability:

The raw data are available as Tables S1 and S3. Data resulting from analyses are presented in Tables 1 and 2 as well as Tables S4–S9 and in all figure files (including supplemental).

The raw data are available in the Supplemental Files. To maintain confidentiality of fossil localities in accordance with community practice, precise locality data for each specimen are on file at the appropriate institutions.

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
