# Peer review of "Using GIS to examine biogeographic and macroevolutionary patterns in some late Paleozoic cephalopods from the North American Midcontinent Sea"

_PeerJ, doi:10.7717/peerj.6910_

## Round 0.1 · original submission · Major Revisions

Thank you for having submitting this interesting and important manuscript entitled “Using GIS to examine biogeographic and macroevolutionary patterns in late Paleozoic cephalopods from the North American Midcontinent Sea”, to PeerJ.

We have now three review reports that consider your investigation very relevant and agree that it would deserve to be accepted for publication in PeerJ after to revise, improve and explain with more detail some of your results. Particularly, there were concerns among the reviewers about your interpretation of data shown in figure 3. While this figure suggests a dramatic increase of the geographic range of cephalopod distribution, you just did not find significant statistical values that support that peaks. Something may be wrong there. I had further concern related to the result, because how is possible that you have a real increased geographic range during a stage (Virgilian) where you say that found the lowest speciation values (S). Maybe the increased range observed corresponds to an invasion of taxa to the basin (a possibility that you mention in the text) or maybe you are assisting to a taphonomic event of increased, sudden mortality that is showing the real distribution of the group in the whole basin?. Another explanation could be that one of your results is wrong and should be revised.

Other important concerns are well explained by the reviewers and are essential to address too (e.g., the differential effects of glaciations on the pelagic/benthic taxa and the number of species used in this study and the accuracy of the previous data on species distribution in which you based your calculations).Note that if your basic data are not correct you can be making wrong assumptions or elaborating not guaranteed hypotheses.
Thus please, revise and fix all the issues that generated the reviewer concerns, even those included in the annotated pdf versions that the reviewers kindly prepared for you to follow.

Once you complete these requests, submit the revised version of your manuscript for a new consideration, I cannot discard the possibility of a complementary second revision, if needed.

I would like to very thank Reviewer 3 for his effort to complete the review of this manuscript even the health problems that he had through. I am happy to know that all is okay with him and his family now. Also, I offer my excuses to the authors for the delay in the review process of this manuscript.

Hopping to see your improved manuscript very soon, I send you my best wishes.
Graciela Piñeiro

·

Basic reporting

The study is doubtlessly interesting as it focuses on a question underrepresented in cephalopod palaeontology. It may work as a case study for including GIS in the study of evolutionary patterns in small scale, i.e. within-basin distributions of species. The metods appear to be solid.
However, there are some question to be asked with respect to the manuscript.
The methods and their historical aspects are discussed in great detail with numerous citations. Is this really necessary? The length of this part leaves the impression that the manuscript is a review article. Contrary, almost nothing is said about the studied objects, the cephalopods. Of the few cited articles, most are dealing with nautiloids. It is really not clear what material and literature was used.

Experimental design

The more serious point is that is not clear from where some of the used numbers come. There is no list of species in stages, from which the numbers from species richness can be obtained. I would strongly recommend to add this in the supplementary material.
A check of table S3 led to the result that there are severe mistakes, which most probably caused major errors in the analyses. Remarkable in this table is:
a) there are only 23 ammonoid species – is this sufficient for a thorough analysis?
b) according to the table, 13 of the 23 species are singletons
c) according to my knowledge, for 9 of the 23 species, incorrect distributions are given
d) six of the 23 species are shown as long-ranging are in fact short-ranging, restricted to the Missourian and Virgilian
It is not explained why the museum-specimen-method is used, which requires a lot of knowledge and experience. Why has not the literature itself been used as the primary data source? Why hasn’t the Paleobiology Database used for the nautiloids and the AMMON Database for the ammomoids? In the latter, there are vetted data available in better stratigraphic resolution and updated taxonomy. It even provides geographic positions.

Validity of the findings

Summing up the points above, I really do not think that the results are sound. There must be doubts if the results, when using a better data base, would be the same.

Additional comments

I am really not happy for the very critical remarks and would like to see the paper published (of course, using better data). The authors may feel free to contact me, as I would like to contribute some constructive items to balance the destructive review.
Comments on details can be found in the annonated manuscript file.

·

Basic reporting

This excellent manuscript analyzes the temporal patterns of diversity, extinction, origination, and geographic range size of ammonoids and nautiloids in the Midcontinent USA. The writing was clear and unambiguous, and the authors appropriately contextualized their work with a sufficient background review of the relevant literature. Raw data were provided in the form of museum specimen codes, although the specimen list did not include geographic coordinates or ages that were assigned to each occurrence. Consequently, I was not able to repeat the analyses. The data on geographic ranges (Table S3) shows many repeated occurrences of a single value (78.539816)—93 of the 170 non-zero values—which suggests that a few collections might be dominating the pattern.

Experimental design

The strongest aspect of this study, to me, is that it fills a gap in a clearly-defined and important research question. The apparently unusual behavior of the biota during the Late Paleozoic Ice Age has been apparent since Sepkoski published his analyses of extinction and origination rates from his compendium. Despite clarification and amplification of the basic pattern since that time, there still remains several key issues, among them that the pattern has mostly been documented at higher-than-species taxonomic level and for the benthic marine fauna or terrestrial flora. Benthic and pelagic diversity patterns in the present-day marine realm are wildly dissimilar (e.g. OBIS data, http://www.iobis.org/), so it stands to reason that pelagic taxa during the Late Paleozoic may also show divergent diversity patterns from the benthic invertebrates—heretofore we have not had a species-level study of a pelagic taxon for a specific region. This study both increases the resolution and expands our understanding of the biotic effects of the ice age. The methods are clearly described and I had no trouble following the analysis. The authors have been careful to consider alternative explanations or confounding effects, e.g., by considering the effect of geographic ranges that extend outside the midcontinent, and by calculating speciation/invasion and extinction/extirpation rates with and without singletons (lines 259ff).

I did wonder about the impact of sampling and think that the authors should do more to address this potential issue. They did analyze the effect of outcrop area, but the problem of variable sampling intensity goes beyond that. The basic pattern of cephalopod diversity (number of species) is a convex-downward pattern with the apex during the Desmoinesian (the overall pattern is dominated by nautiloids). But unless sampling is complete (100% of species have been collected and identified), the true diversity pattern needs to be estimated with a different metric and might be different. I couldn’t re-analyze the authors’ data, but I downloaded stage-level occurrences of Laurentian cephalopod genera from fossilworks.org and calculated SQS and Fisher’s alpha (FA), two commonly used methods of sample-standardizing diversity data. The data are not equivalent to the authors’, so they should not be expected to show the same pattern, but they do illustrate that sample-standardized patterns might be very different than raw number of species (S):

ICS Stage S N SQS FA
Sakm 35 77 12.2 25.1
Asse 24 61 6.8 14.8
Gzhe 38 158 7.2 15.9
Kasi 50 131 12.6 29.7
Mosc 48 286 7.3 16.5
Bash 54 422 8.3 16.5

Similarly, the number of species can effect geographic range sizes. If more species are sampled in certain intervals, that might tend to either decrease mean geographic range size (because more small-range species were sampled) or increase it (if the sampling area increased, then the probability of finding range edges increases). Some type of sampling-standardization on geographic ranges would help distinguish whether these factors mattered.

The authors should also include error bars on geographic ranges (not included in the figures) so that significance of changes could be assessed more readily.

Validity of the findings

The data are sound and the conclusions are clearly stated and connected to the original question. The authors have not over-interpreted their results (for example, they were careful when discussing glacial cycles that may have occurred on timescales within temporal bins, which their data would not show). Their choice to separately analyze ammonoids and nautiloids was, to me, particularly helpful in assessing my confidence in their results. Assuming that the issues I raised above do not substantially affect the results, then I see no reason why their findings are not valid.

Additional comments

Some random, minor comments.
a. Figs S1-S3 could be plotted on same graph so can be compared.
b. Line 217 is missing a word, should be speciation rate [or] extinction rate.
c. I am not surprised about the weak statistical relationship between rates and geographic range, because all comparisons are for species within the ice age, i.e., a culled subset following the late Mississippian extinction.
d. Table 2 shows very high correlation coefficients…given the sample sizes, I am surprised that they could be so high and yet p-values just marginal or not statistically significant. For example, the correlation between geographic range and extinction for all cephalopods is r = 0.92, p = 0.075! Can the authors double-check this, or am I misinterpreting something?
e. In Table S3 I was confused about whether a “0” value meant that the species did not occur in that time interval or if it was a single occurrence. The authors stated that a cushion was used around a single occurrence so I expected a non-zero value. But maybe I just missed this explanation in the text.

·

Basic reporting

The report is very well written and appropriately referenced. I have made some comments about a very small number of minor issues regarding sentence structure and the use of unfamiliar vocabulary. However, the authors can ignore this advice if they see fit as it is entirely superficial to the study. I think some of the figures could be more informative - see general comment section for line by line breakdown.

Experimental design

The study is well designed and does a thorough job of answering the questions set out in the introduction. The authors use appropriate and established methods to assess biogeographic range, various evolutionary rates and to test for biases in the fossil record.

Validity of the findings

I am little concerned by the drastic change in range size in the Virgilian as illustrated in Figure 3 s it do not feel this is highlighted enough in the text and I suspect the non-significance of the result is influenced by the method used as the plot clearly shows a dramatic increase (particularly in the nautiloids) from close to zero to >6x10^8 (what I'm guessing is km^2). I'd, therefore, like to see a more in depth explanation of the results and why they're being dismissed as non-significant changes in range size when the plots seems to suggest otherwise. So, to summarize, even though range size doesn't appear to correlate with anything else, the paper reads as though there are no significant changes is range size through the period of interest, a result that is clearly contradicted by the plots in figure 3.

Other than this issue, I like the rest of the study and think the findings are deserving of publication.

Additional comments

Thank you for giving me the opportunity to review this manuscript and I thoroughly enjoyed reading it. I apologize to the authors and the editor for the tardiness of this review as I realize I am a few weeks behind schedule. I have been hampered by illness and ill children over the past few weeks and am only just getting through my backlog of things to do.

I firmly support the publication of this manuscript once the single major issue outlined in 3. is addressed and after the authors consider my list of minor points below:

Lines 42-44: this covers the issue highlighted in 3 but also I feel there could be more explanation (in the manuscript in general) about why we expect pelagic organisms to behave differently to benthics. This issue is repeatedly touched upon but I want more reasoning.

Line 67: is this ever the case for ammonoids? They're the textbook group for high turnover and rapid evolution.

Line 74: I'd incorporate the sentence about fusulinids into the previous sentence about crinoids - as it is, it seems a bit lost on the end of the paragraph.

Paragraph starting line 78: I haven't read the references but wonder why and how environmental change would be cited a reasons for sluggish rates? I would have assumed the opposite i.e. environmental stasis = sluggish evo rates. Perhaps elaborate on the reasoning these refs give as to this conclusion in order to give the reader a little more context.

around line 86 onwards: is it fair to say that pelagic taxa might well be less influenced by glacial sea level cycles as these events are also known to cause variation in seafloor ventilation - thus any dysoxia/anoxia would be more severe for benthic communities that pelagic faunas?

Paragraph starting line 109: Would the sea level change have significantly altered the area of the midcontinent sea? If so, then surely it would also affect range size of pelagic organisms within the basin? Even though these changes would perhaps be less problematic for pelagic organisms than depth-dependent benthic communities, I would still expect it to drive biogeographic patterns. Is it possible to estimate the area of the sea throughout this study period? If so, it would be nice to plot this in ArcGIS to get a good visual representation.

Line 133 onwards: how do your spatial patterns from the museum collections compare with PBDB data? Does it show that you're missing any data or vice versa, does it show that PBDB is missing data i.e. unpublished museum records. The former would be a problem for this study whilst the latter would be interesting in light of this recent article: http://rsbl.royalsocietypublishing.org/content/14/9/20180431?casa_token=7cBBG9-o7SMAAAAA%3AT4aO4AwdEyx1fk7A86VRj-z0rGDhNhhLhELyQYrLQIXfMyuzmatvd1-6-QF6KJVWmQn_DWRh4yzN0bp0

Line 188: I had to look up "cognizant". It might be that I'm not that bright, but I'd change this to something more widely understood like "aware".

Line 201 onwards: If I understand this reasoning correctly, could it not also be that a good correlation between outcrop and range size means that cephalopods were just occupying the full extent of the midcontinent sea and would always closely match the outcrop area of midcontinent sea sediments? Outcrop would limit range in this case but it will only be a problem if outcrop area varies significantly through time. Therefore, I'd be concerned if both outcrop area AND range size shows a large increase in the Virgilian. I think the authors need to clarify whether this is the case before they make the claims that outcrop does not drive range size.

Line 205: Quite but see Dunhill et al. 2014 for some solid evidence that outcrop area does likely drive diversity (and thus probably also biogeographic patterns) in the marine realm, if not the terrestrial. https://www.nature.com/articles/ncomms5818

Lines 24 onwards: what do mean by loosely correlates? Significant? This statement makes sense and should be emphasized more in the discussion as the range size increase during this time is huge.

Lines 320 onwards: I'm a little uncomfortable with almost significant.. also are these positive or negative correlations?

Discussion: I think the discussion needs some addition following the suggested clarifications listed above - i.e. more credence given to massive range increase around the Virgilian and a second look at the outcrop vs range relationship. Don't just rely on correlation as the sample size is small.

Figures:
Figure 1: more detail please - lat-long graticule, scale etc, N arrow - all standard for map publication.

Figure 2: as above. Also, can we zoom in here? it's really hard to make it out at this scale - you could do a two panel map with a small global extent with a zoom square and then the zoomed in portion much larger. Quite easy to do in Arc. Also, would it be possible to map the extent of the midcontinent seaway on here? Or at least the outcrop of midcontinent seaway sediments?

Figure 3: This is where we see the huge extent of range change in the Virgilian - needs more making of this throughout the manuscript. I think the y axis would be better displayed in 10^ rather than the long list of zeros. Also, what is the unit? I'm guessing km^2?

Figure 4: Again, what is the unit here? n species per Ma?

I'm happy to see any revised versions of this manuscript but I'm also happy for the editor to accept based on the addressing of my single main concern and the list herein.

Sorry again for the lateness of this review and best wishes with the revisions.

Alex Dunhill
University of Leeds, UK

---

## Round 0.2 · Minor Revisions

Dear authors,

Regarding the concerns described in the revision of the last version of your manuscript, either from one of the reviewers and from my own (see please, the attached annotated PDF), I decided to send the article for an independent new review. The review report from the new reviewer is now available and it seems to have found very coincident weakness in the interpretation of the data, remarking the possibility that these weaknesses are mainly due to the small sample size and the very long time bins selected. I think that your manuscript is a very important contribution to a better understanding of how much the variation of climatic conditions may have affected marine communities in the geological past, particularly referring to Carboniferous pelagic groups. But as such, it needs to be based on the most part of available evidence and reliable (easy verifiable) data. The new review (which I appreciate so much) is a very good compilation of the troubles that specialists in the matter would detect when reading the current manuscript, but the fixing of them would transform this article in a very useful study for a wider scope. Thus, please, pay attention to any of the reviewer recommendations and concerns, some of them can be easily fixed and others will rather require a little more work; and take also into account my own recommendations in the attached annotated PDF. Once you have the new version of your manuscript, please submit it for a new revision by the editorial staff. Thank you very much in advance.

With my best wishes,
Graciela Piñeiro

·

Basic reporting

Comments on the manuscript by Kolis and Lieberman

“However, we feel it is very important to note that it is a well-known phenomenon that data in the PBDB are often not necessarily accurate.”
- The authors are completely right regarding the PBDP. But what about AMMON? In that case all data have been vetted. One cannot state that both databases, which were made for differing approaches, are similar in their data quality.
“What we are doing here is presenting a hypothesis on the stratigraphic and geographic distribution of Pennsylvanian-Permian cephalopods from this part of North America.”
- Basically, there are already data for the stratigraphic and geographic distribution of Pennsylvanian-Permian cephalopods” – mainly collected in the literature on the objects.
“For Dieter Korn to say that our result must be “wrong” simply because of that does not seem fair to me.”
- We can strictly separate between data and hypotheses build on data. The presence and occurrence of specimens is data. When this is only hypothetical, where is then the data? This would mean that hypotheses are built on hypotheses, in my opinion the wrong approach.
“I feel the best way to treat our result is as a hypothesis; that’s all we’re trying to do.”
- No problem with it, but as I said above – I prefer hypotheses based on solid data, not on hypotheses on data.
“We do happen to disagree with Dieter Korn on this aspect of our data. I wouldn’t go so far as to say that “he is wrong” or “our data are better”; essentially that is what he is trying to say about the data he uses.”
- Between the lines, one could get just this impression. However, the data quality can easily be tested. Some examples:
- The authors list ten ammonoid species from the Virgilian. The AMMON database lists 47 species alone from the Virgilian of Texas (data mined from the literature)! Is 10 out of 47 representative?
- The authors list Schistoceras hildrethi from Morrowan to Virgilian. But, the species Paraschistoceras hildrethi was described, from a large number of localities, only from Missourian and Virgilian strata.
- The authors list Schistoceras missouriense from Atokan to Wolfcampian. But, the well-known index species was described, from a large number of localities, only from Missourian and Virgilian strata.
- The authors list Schistoceras unicum from Atokan to Virgilian. But, the species Eochistoceras unicum was described from Missourian localities in Oklahoma.
- One has to ask here what is then the hypothesis? Were all the stratigraphic statements by the North American ammonoid workers incorrect? Do the authors want to challenge the widely accepted idea that Schistoceras is a Missourian-Virgilian index fossil and occurs already long before? If they do this, they try to refute the Carboniferous ammonoid stratigraphy as a whole. But to do this, one would require very solid data.
“We have built these data up from years of consideration of the actual specimens themselves, both as figured in the literature and also as housed in museums.”
- But who is responsible for the identifications and locality statements?
“If Dieter Korn wants to present a different hypothesis in a paper that he writes of course that is fine, though we might disagree with aspects of his hypothesis about distributions as well.”
- I find this is a very strange point. Of course I could write an own paper. But this is not to discuss here, but the actual manuscript.
- To make it clear: I do not speak about “my hypotheses” about the ammonoid distributions. Apart from some short collection visits, I did not work on the Pennsylvanian ammonoids of Texas and Oklahoma. These are not my data. I speak about data collected by the earlier palaeontologists (among which are very good workers like Miller and Furnish) in that region, AMMON just contains these data. I was only the data collector; the “hypothesis on the stratigraphic and geographic distribution” by the authors is largely expressing doubts on the basis of their research – the correct identification of ammonoid specimens.
- I do not wish to be misunderstood. Challenging hypotheses and theories is, not only according to T.S. Kuhn, the best scientific practice. Knowledge about the Signor-Lipps effect, for instance, helps a lot. Calculating confidence intervals for the temporal distribution of species is necessary. But the basis must be solid data.

Experimental design

-

Validity of the findings

-

·

Basic reporting

The manuscript is well written and in most parts easy to understand.
The introduction includes an extensive review of previous papers that deal with the general relationships between geographic range, speciation and extinction, but it does not pay much attention to specific ammonoid literature that dealt in the past with questions of Pennsylvanian/Permian ammonoid palaeodiversity and the relationships between glacio-eustatics and ammonoid evolution (e.g. various papers by Ramsbottom, Kullmann…). There should be a short paragraph how cephalopod evolution and palaeodiversity of the investigated time interval have been viewed in previous publications. This will widen the background of the study.
The quoted literature is extensive. The reference chapter misses (only) two papers cited in the text: Jacobs et al., 1994 and Miller & Owens, 1934.
The manuscript has overall a clear and good structure. As a minor point, some of the applied statistical techniques are not explained in the methods paragraph although they are mentioned in the subsequent text. Short explanations will improve the manuscript readability and it easier to understand for a wider audience.
The figures are relevant but some could be improved. Fig. 1 shows the study region but in the absence of state borders, the individual dots are difficult to align with published faunas and specific regions. In Fig. 2, the precise position of palaeolatitudes should be shown by lines. There should be an additional figure that gives an example how the reconstructed spatial distribution of a selected taxon plots in comparison with the Midcontinent palaeogeography of the relevant time bin.
The supplement figures and tables are essential and very helpful.

Experimental design

The manuscript reports interesting original primary research.
The research question is clearly formulated and addresses, based on well suited case study, a field of palaeontology that is of wide interest and in the focus of other ongoing research.
It applies the most modern statistical and geospatial techniques. The methods are, however, partially rather briefly explained (see above); their full understanding requires own experience in the field.
A principle problem of the study is the almost random selection of a small part of the much more diverse total cephalopod faunas of the Midcontinent. In addition, the chosen time bins (stages) are very (too) long. This questions how representative results are. However, it cannot be denied that the study fills a gap of knowledge and it can serve as a base for future investigations.

Validity of the findings

As noted above, the data base is far from complete and its conclusiveness will have to stand the test of time. The possible enlargement would probably require several years of additional work. Therefore, the results have to be seen as a first step towards a future bigger and more detailed view, especially with a higher time-resolution. This aspect is clearly outlined in the discussion chapter.
The statistical handling of the used data is mostly fine, although the questionable partial exclusion of singleton taxa should be better explained and critically reflected.
The conclusion are well linked with the data and original research question. Statistical uncertainties are clearly explained. There is not too much speculation – the interpretation remains rather conservative and close to the data and analyses.

Additional comments

1. This seems to be a too simplified assumption, especially with respect to cephalopods. Since Ramsbottom (1980, in House & Senior, eds., “The Ammonoidea”), which, unfortunately, has not been considered, the rapid glacio-eustatic cycles of the Upper Carboniferous are known to have fundamentally influenced ammonoid diversity and distribution in space and time, with many taxa being confined to single glacial cycles. This patterns is known from several continents (North America, Europe, the Urals). The duration of cycles varied but it is clear that glacio-eustatics was a main driver of speciation and extinction. The authors should incorporate this classical view into their introduction although it was mostly based on examples that are somewhat older than the study time interval.
2. In this context you should also consider the palaeoecological constraints that reduce the dispersal potential of cephalopods. Most require a minimum water depth for vertical migration. Therefore, their distribution is strongly restricted by too shallow palaeogeographical barriers (seaways) and sea-level falls caused regular and repeated, partly very strong, extinction. A summary of palaeoecological constraints for ammonoid distribution in the mid-West was also given by Mapes (2004): Field Trip Guidebook for the 6th International Cephalopod Symposium.
3. This is too simplified because it neglects the highly significant aspect of sea-level fall, which is widely known to have caused ammonoid extinctions. In additions, many Palaeozoic cephalopods were sensitive to water temperature, which explains their rarity and low diversity in the high palaeolatitudes.
4. Unfortunately, this appears to be a random selection of taxa, probably stimulated by the studied collections of the three named universities. The total diversity of ammonoid species in the region is much much higher. Good preservation, clear taxonomic status and abundance apply to many other taxa that were not included. For example, I had no problems to collect in 2004 many of the species that have been described from the Finis Shale of northern Texas. Since it is obviously not possible without several more years of work to cover all taxa (which would be the best), the authors must make a better statement why the selected taxa are representative enough to enable general conclusions. This is especially important for the evaluation of the speciation, extinction, turnover, and total diversity rates, which will be very different for the total faunas. I am very sympathetic towards the museum specimen approach but the total number of 1.100 specimens is not very high with respect to the fact that it is easily possible to collect more than hundred goniatites in a day at some localities. I agree that the PBDB is not suitable (still far too incomplete) for the scope of the present study.
In the table of all analyzed taxa, “Orthoceras” rather than Orthoceras should be used. The true Orthoceras is a lower Palaeozoic genus; all Carboniferous-Permian species will have to placed in different genera and they may belong either to the Palaeocephalopoda (Pseudorthocerida) or Neocephalopoda (Orthocerida).
5. Since the region has such a detailed ammonoid zonation, it would have been much more interesting to use the zonal scale as the base for analyses. Please explain why this was not considered. Many speciations and extinctions occurred within the stages, as did sea-level changes.
6. It would be good to visualize this on a map for one or two selected taxa, not necessarily in the appendix. It would be especially important to know how the created range fits to the relevant palaeogeographic map for the age of the selected taxon.
7. There is currently no other way to do this but you should perhaps raise the problem that the absolute position of boundaries of the international stages is based on interpolation between still too few geochronological tie points and that the correlation of the North American stage boundaries with these is arbitrary in addition, which adds further uncertainty. The reader should be aware that the used upper boundary ages are subject to further research (and will certainly change: the chapter for the revised GTS 2020 should have been submitted at the end of 2018).
8. This part should go into the Methods chapter and you should try to explain the “Anderson-Darling normality text” (references?).
9. Which taphonomic condition could increase the preservation of cephalopods? They can be variably preserved (3D) in limestones, as 3D-moulds in black shales, or squashed on shale surfaces (if there was not sufficient bacterial sulphate reduction for pyrite/marcasite formation). In the latter case, such poorly preserved specimens may not have been collected (therefore, a collecting bias, not a taphonomic bias). The “taphonomic bias” does not sound likely and would require some additional explanation.
10. The Mann-Whitney U test has to be explained, perhaps with a reference, in the Method chapter – it is not necessarily known to the majority of readers.
11. Please mention and briefly explain the purpose of the t-test in the methods chapter.
12. I wonder that there is no reference to the overall and changing palaeogeographic setting of the basin. Did the basin widen/expand during the Virgilian?
13. Since ammonoids are commonly (see your table) short-ranging and often to mostly restricted to single stages or even to single biozones, the exclusion of singletons needs to be justified here (or better in the methods chapter). For ammonoids, there is no reason to assume that singletons carry a higher bias than longer-ranging species. I cannot judge whether this is also true for Carboniferous/Permian nautiloids.
14. Therefore, there should be in future a zone by zone analysis for all known cephalopod taxa from the Midcontinent.
15. In the cases that genera have a lower range outside the Midcontinent it should be clear that there were immigration events. Are there examples?
16. But in the case of ammonoids, spreads with eustatic rise are more important than temperature changes (e.g. Hartenfels & Becker 2016 for the global Annulata Events in the Famennian, Zhang et al. 2018 for the Hangenberg Black Shale at the D/C boundary). Ammonoids required wide outer shelf ramps and basins; shelves became too narrow and shallow during regressions.
17. But it should be possible to link the distribution patterns of taxa within individual biozones with the glacio-eustatic pulses that (back to Ramsbottom 1980).
18. This is probably the important point: you have to look at a finer stratigraphic scale that directly can link biozones and individual glacio-eustatic events, which are easily manifested sedimentologically.
19. In this context you should perhaps refer to some of the publications that deal with global ammonoid diversity in the Upper Carboniferous-Lower Permian, such as Kullmann (1983, 2002, 2007), Wiedmann & Kullmann (1996) or Becker & Kullmann (1996).

---

## Round 0.3 · Minor Revisions

Dear authors,

Thank you for to have submitted a revised version of your article on Late Paleozoic cephalopod biogeography and macroevolutionary patterns. I am glad to see that you followed the useful and relevant recommendations from the last reviewer, Dr. Ralph T. Becker and modified accordingly your manuscript; indeed, it was notably improved by adding these modifications. I have just concerns about your explanation on the exclusion of singletons, it is too much confuse and this is an important fact that can even influence your results. Thus, please see my comments in the annotated pdf that is being attached to this letter and fix these paragraphs. The pdf file also remarks other issues that need to be addressed and hopefully, you will consider them useful and can submit a revised new version of your manuscript very soon.

With my best regards,
Graciela Piñeiro

---

## Round 0.4 · accepted · Accept

Dear authors,
Thank you for your submission and for having considered the last recommendations. I have to be honest to say that I had serious concerns about the destiny of this contribution when reading the earliest versions of this manuscript. Even though, the input that you received fundamentally from the reviewers has transformed substantially the content of your study, making it more reliable although not definitive, without alteration of its proper essence. I am very glad and grateful to the reviewers that collaborated in this long process. That said, I am happy to announce you that your manuscript is ready for publication in PeerJ. Congrats!
With my best regards,
Graciela Piñeiro

# #